# BrightDreamer: Generic 3D Gaussian Generative Framework for Fast Text-to-3D Synthesis

**Lutao Jiang**  *ljiang553@connect.hkust-gz.edu.cn*
*Thrust of Artificial Intelligence*
*The Hong Kong University of Science and Technology (Guangzhou)*

**Xu Zheng**  *xzheng287@connect.hkust-gz.edu.cn*
*Thrust of Artificial Intelligence*
*The Hong Kong University of Science and Technology (Guangzhou)*

**Yuanhuiyi Lyu**  *ylyu650@connect.hkust-gz.edu.cn*
*Thrust of Artificial Intelligence*
*The Hong Kong University of Science and Technology (Guangzhou)*

**Jiazhou Zhou**  *jzhou297@connect.hkust-gz.edu.cn*
*Thrust of Artificial Intelligence*
*The Hong Kong University of Science and Technology (Guangzhou)*

**Lin Wang**[†]  *linwang@ntu.edu.sg*
*School of Electrical and Electronic Engineering*
*Nanyang Technological University*

**Reviewed on OpenReview:** *https://openreview.net/forum?id=Rb19CQCwbi*

## Abstract

Text-to-3D synthesis has recently seen intriguing advances by combining the text-to-image priors with 3D representation methods, *e.g.*, 3D Gaussian Splatting (3D GS), via Score Distillation Sampling (SDS). However, a hurdle of existing methods is the low efficiency, per-prompt optimization for a single 3D object. Therefore, it is imperative for a paradigm shift from per-prompt optimization to feed-forward generation for any unseen text prompts, which yet remains challenging. An obstacle is *how to directly generate a set of millions of 3D Gaussians to represent a 3D object*. This paper presents ***BrightDreamer***, an end-to-end feed-forward approach that can achieve generalizable and fast (**77 ms**) text-to-3D generation. **Our key idea** is to formulate the generation process as estimating the 3D deformation from an anchor shape with predefined positions. For this, we first propose a Text-guided Shape Deformation (TSD) network to predict the deformed shape and its new positions, used as the centers (one attribute) of 3D Gaussians. To estimate the other four attributes (*i.e.*, scaling, rotation, opacity, and SH), we then design a novel Text-guided Triplane Generator (TTG) to generate a triplane representation for a 3D object. The center of each Gaussian enables us to transform the spatial feature into the four attributes. The generated 3D Gaussians can be finally rendered at **705 frames per second**. Extensive experiments demonstrate the superiority of our method over existing methods. Also, BrightDreamer possesses a strong semantic understanding capability even for complex text prompts. *The code is available in the supplementary materials.*

---

[†] Corresponding author.

# 1 Introduction

Text-to-3D generation has received considerable attention in the computer graphics and vision community owing to its immersive potential across diverse applications, such as virtual reality and video gaming (Li et al., 2023a). Recently, with the emergence of diffusion models (Ho et al., 2020; Rombach et al., 2022) and neural rendering techniques (Mildenhall et al., 2021; Kerbl et al., 2023), text-to-3D has witnessed an unprecedented technical advancement. In particular, pioneering methods, such as DreamFusion (Poole et al., 2022), LatentNeRF (Metzer et al., 2023), SJC (Wang et al., 2023a), have sparked significant interest in the research community, catalyzing a trend toward developing techniques for creating 3D assets from texts. The follow-up methods then focus on either quality improvement (Raj et al., 2023; Shi et al., 2023; Wang et al., 2023b; Liang et al., 2024) or geometry refinement (Chen et al., 2023a; Lin et al., 2023) or training efficiency (Tang et al., 2023; Yi et al., 2023).

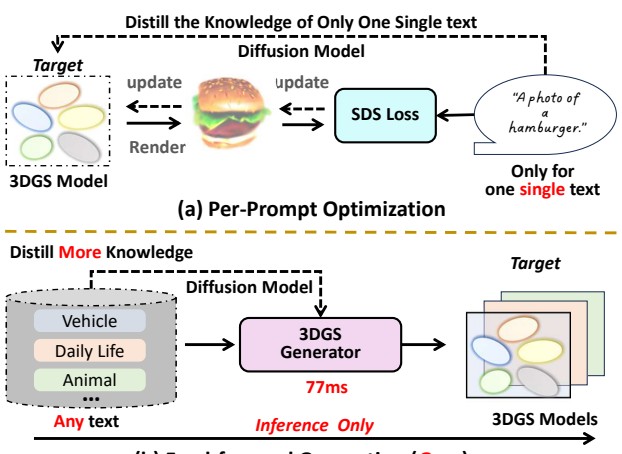

Figure 1: A comparison between per-prompt optimization-based methods, and our feed-forward generation-based approach with an end-to-end objective.

The dominant paradigm of these methods is to randomly initialize a 3D representation model, *e.g.*, Neural Radiance Fields (NeRF) (Mildenhall et al., 2021) or 3D Gaussian Splatting (3D GS) (Kerbl et al., 2023), and optimize such a model to align with a specific text prompt, as depicted in Fig. 1 (a). Unfortunately, these methods suffer from two critical constraints. ***Firstly***, as per-prompt optimization usually requires several tens of thousands of iterations, this inefficiency brings a considerable obstacle to broader applications. It is significantly different from the mainstream training paradigm in the field of 2D image generation (Song et al., 2020; Rombach et al., 2022; Ramesh et al., 2022) or 3D-aware image generation (Schwarz et al., 2020; Chan et al., 2021; Jiang et al., 2023; Or-El et al., 2022; Chan et al., 2022): *a generative model is trained with a collection of text-image pairs or images, and the model can generate the desired content from any input at the inference stage.* ***Secondly***, as shown in Fig. 2(a), existing methods often fail to accurately process the complex texts. For example, the mainstream methods struggle in generating 3D content that input prompt contains complex interaction between multiple entities. This limitation arises from the models being trained on a single text prompt, resulting in a degraded capability in comprehensive semantic understanding.

Therefore, it is urgently needed for a paradigm shift from per-prompt optimization to develop a *generic* text-to-3D generation framework. Once trained, the generative framework should be able to generate content from any text prompts in the inference stage, as depicted in Fig. 1(b). Furthermore, given the **scarcity of 3D data** in comparison to the abundance of 2D image data, leveraging well-trained 2D image diffusion models to facilitate the training of 3D generative models presents a more effective and resource-efficient approach. Previously, some research efforts, *e.g.*, ATT3D (Lorraine et al., 2023) and Instant3D (Li et al., 2024c), have been made grounded in NeRF representation. The core insight is to add a large number of texts and take them as conditional inputs to generate explicit spatial representations, such as triplane (Chan et al., 2022). Nonetheless, in stark contrast to the volume rendering in NeRF, 3D GS representation for an object usually consists of millions of 3D Gaussians. Consequently, there exists an inherent and natural difficulty in converting the generation representations into 3D GS ones in their framework.

In this paper, we propose ***BrightDreamer***, an *end-to-end feed-forward* framework that, for the **first** time, can achieve generalizable and fast (77 ms) text-to-3D GS generation. BrightDreamer exhibits a robust ability for complex semantic understanding (Fig. 2 (a)), and it demonstrates a substantial capacity for generalization (Fig. 2 (b)). Additionally, like traditional generative models (Goodfellow et al., 2014), our generator can interpolate between two inputs (Fig. 2 (c)), allowing users to unleash their imagination and creativity, thus expanding the potential for novel and nuanced design exploration. As stated before, the

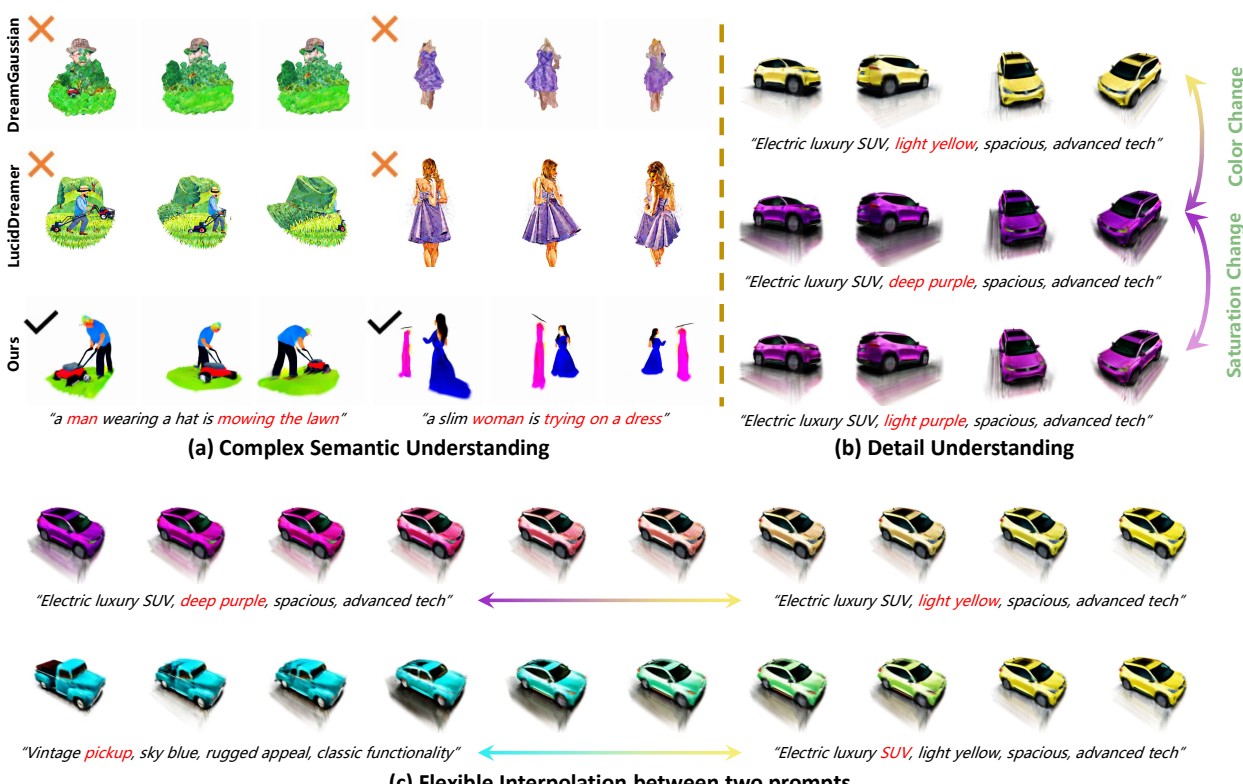

Figure 2: DreamGaussian (Tang et al., 2023) and LucidDreamer (Liang et al., 2024) are both optimized for a single text. Our result is the direct generation. And for the display of our generalization, **all the prompts do not appear in our training set**. (a) is for showing the complex text understanding. (b) is to demonstrate our capability of understanding details. It is noteworthy that **light purple**, **deep purple**, and **light yellow** don't appear in the training set. (c) Interpolation between two prompts from color and shape perspectives.

3D GS representation of an object usually comprises several hundreds of thousands of 3D GS elements. Thus, directly generating such a large collection is impractical. **Our key idea** is to redefine this generation problem as its equal problem, *i.e.*, 3D shape deformation. Specifically, we place and fix some anchor positions to form the initial shape. Then, it can be deformed to the desired shape by giving different input prompts through our designed Text-guided Shape Deformation (**TSD**) network (Sec. 3.1). Then, the new positions can be set to the centers of the 3D Gaussian. Upon establishing the basic shape, we elaborately design a Text-guided Triplane Generator (**TTG**) to generate a spatial representation of the 3D object (Sec. 3.2). Subsequently, we utilize the spatial feature of each center of 3D Gaussian to represent its feature and translate it into the remaining attributes (including scaling, rotation, opacity, and SH coefficient) through our well-designed Gaussian Decoder (Sec. 3.3). For TTG, based on our re-analysis of the previous convolution-based triplane generation process, we identified and solved two primary deficiencies. One issue involves the spatial inhomogeneity observed during the calculation process, as shown in Fig. 4. The other issue arises from the single-vector style control mechanism similar to StyleGAN (Karras et al., 2019), which complicates managing relationships between multiple entities.

Our contributions can be summarized as follows: (**I**) We propose BrightDreamer, the first 3D Gaussian generative framework to achieve generalizable and fast text-to-3D synthesis. (**II**) We design the Text-guided Shape Deformation (TSD) network to simplify the difficulty of direct generation of 3D Gaussians. We design the Text-guided Triplane Generator (TTG) to generate the object's spatial features and then decode them as the 3D Gaussians. For TTG design, we re-analyze and solve the existing problems in the mainstream triplane generator, including spatial inhomogeneity and text understanding problems. (**III**) Extensive experiments

demonstrate that BrightDreamer not only can understand the complex semantics (while the per-prompt optimization methods fail) but also can utilize its generalization capability to achieve generation control.

## 2 Related Works

**Text-to-3D Generation.** Existing methods can be grouped into two categories. **1) Optimization-based methods** typically commence with a randomly initialized 3D model, such as NeRF (Mildenhall et al., 2021), and subsequently employ text-image priors (Radford et al., 2021; Rombach et al., 2022) to guide and optimize its parameters. After undergoing thousands of iterative refinements, this predefined 3D model progressively morphs to embody the shape described by the corresponding text input. DreamField (Jain et al., 2022) represents the inaugural foray into text-to-3D methodology, utilizing the pre-trained text-image model, CLIP (Radford et al., 2021), as a guiding mechanism for the optimization process of a predefined NeRF model. DreamFusion (Poole et al., 2022) proposes the Score Distillation Sampling (SDS) to transfer the prior of the 2D diffusion model (Ho et al., 2020) into a 3D representation model (Mildenhall et al., 2021; Müller et al., 2022), which achieves impressive performance and ignites the research enthusiastic for the text-to-3D task. Inspired by this, numerous works (Wang et al., 2023b; 2024; Liang et al., 2024; Tran et al., 2023; Yu et al.; Li et al., 2025; 2024a; Yang et al., 2024b; Wu et al., 2024) are devoted to re-designing the SDS loss, enabling much more local details of the 3D model. MVDream (Shi et al., 2023) and PerpNeg (Armandpour et al., 2023) attempt to solve the *Janus* problem, *i.e.*, multi-face problem in some text prompts. Though these methods have great generalizability, it usually needs several hours to optimize a 3D model. **2) Generation-based methods**, by contrast, aim to directly generate a 3D model from a given text, streamlining the process of text-to-3D generation. ATT3D (Lorraine et al., 2023) is the first attempt to train a NeRF model with multiple texts by SDS. Though its generalizability is limited, benefitting from SDS, the rendered image is better than previous methods (Sanghi et al., 2022; Chen et al., 2019; Fu et al., 2022; Liu et al., 2022) and the generated content is not limited to 3D data (Mittal et al., 2022; Sanghi et al., 2023). Instant3D (Li et al., 2024c) designs some modules to map the text input to the EG3D model (Chan et al., 2022) and then uses SDS to train this model. Concurrently, there are also some works focusing on text-to-image-to-3D model route (Xu et al., 2024d; Tang et al., 2025; Lu et al., 2024; Wei et al., 2024; Xu et al., 2024b; Wang et al., 2025; Liu et al., 2024; Yang et al., 2024a; Hong et al., 2023; Zou et al., 2024; Li et al.; He et al., 2024; Xu et al., 2024c). However, due to the limited amount of 3D datasets, it is extremely hard to train a generic model to implement this target well (Sec. 4.2.4). Different from the mentioned works, we propose BrightDreamer, a generic framework that, for the **_first_** time, can achieve fast (77 ms) direct text-to-3DGS generation. Moreover, our method doesn't depend on 3D data, which shows a greater potential.

**3D Gaussian Splatting (GS).** Recently, 3D GS (Kerbl et al., 2023) has emerged as the leading method for 3D representation method of 3D objects or scenes, offering faster rendering speeds and greater application potential than NeRF (Mildenhall et al., 2021). In a short time, a large number of methods have been proposed to leverage 3D GS for diverse tasks, *e.g.*, anti-aliasing novel view synthesis (Yu et al., 2023; Yan et al., 2023b), SLAM (Yan et al., 2023a; Keetha et al., 2023; Matsuki et al., 2023; Yugay et al., 2023), human reconstruction (Li et al., 2024b; Moreau et al., 2023; Kocabas et al., 2023; Abdal et al., 2023; Li et al., 2023b; Liu et al., 2023), dynamic scene reconstruction (Luiten et al., 2023; Yang et al., 2023b; Wu et al., 2023; Yang et al., 2023a; Xu et al., 2023), and 3D content generation (Xu et al., 2024a; Chen et al., 2023b; Tang et al., 2023; Yi et al., 2023; Li et al., 2023c; Liang et al., 2024). Our work builds on 3D GS, aiming to develop a generic text-to-3D generation framework that can generate 3D GS with low latency (77 ms).

## 3 Method

**Overview.** BrightDreamer aims to generate 3D Gaussians directly from text prompts. After training, it can generate the 3D GS model with a remarkably low generation latency (about 77ms). And, the generated 3D Gaussians can be rendered at an impressive inference speed of over 700 frames per second. Each 3D Gaussian is defined by five attributes: center $p'$, scaling $S$, rotation $R$, opacity $\alpha$, and SH coefficient $SH$. **Our key idea** is two-fold: **1)** Defining *anchor positions*, *i.e.*, predefined positions, to estimate the center of 3D Gaussians; **2)** Building *implicit spatial representation*, which can be decomposed to estimate the other four attributes of 3D Gaussians.

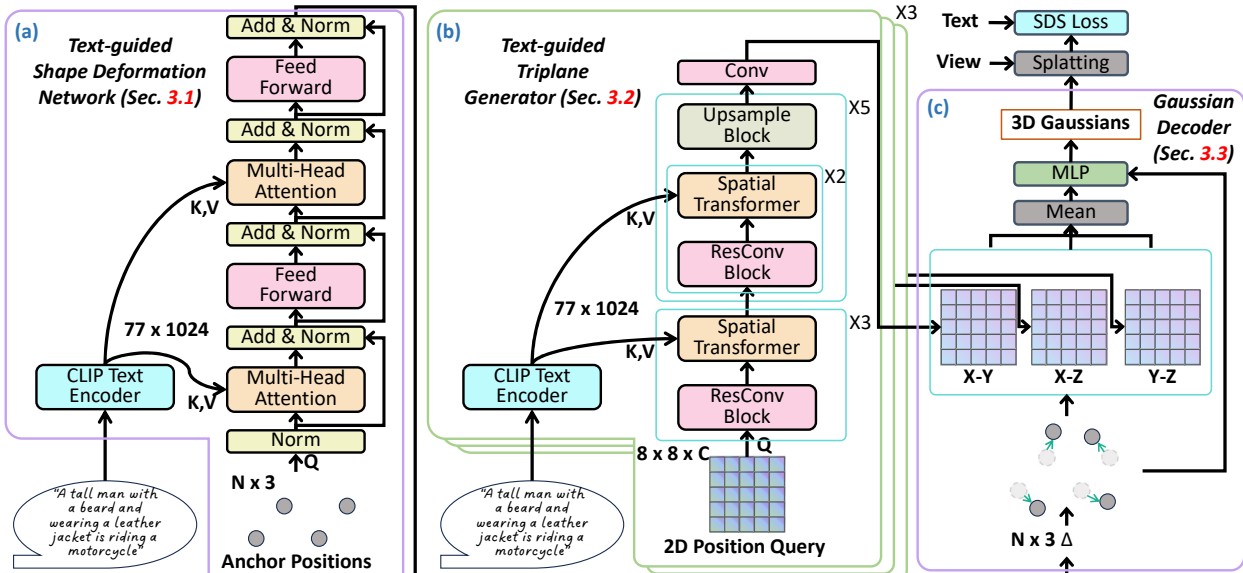

Figure 3: **An overview of BrightDreamer**. The details of Spatial Transformer, ResConv Block and Upsample Block are shown in Fig. 5.

Intuitively, we propose BrightDreamer, and an overview is depicted in Fig. 3. Given a text prompt as input, we transform it to a $77 \times 1024$ embedding through the frozen CLIP text encoder. Next, the TSD network (Sec. 3.1) transforms the fixed anchor positions to the desired shape with text guidance. The new positions are used as the centers of 3D Gaussians. We then design the TTG (Sec. 3.2) to separately generate three feature planes to construct the implicit spatial representation. Based on the centers of Gaussians, we can obtain their spatial features, which are then transferred to the other attributes through the Gaussian Decoder (Sec. 3.3). Finally, we render 3D Gaussians to 2D images and use the SDS Loss (Poole et al., 2022) to optimize the whole framework. We now describe our BrightDreamer in detail.

## 3.1 Text-guided Shape Deformation (TSD)

**The goal of TSD** is to *obtain the center (one attribute) of each 3D Gaussian*. Considering that directly outputting a huge number of center coordinates is extremely difficult, we overcome this hurdle by deforming the anchor positions instead of generating them.

**Anchor Position.** The *anchor positions* are predefined positions, which are the fixed coordinates. It serves as one of the inputs for the TSD. Specifically, we place the anchor positions on the vertices of a 3D grid, as represented by the gray points in Fig. 3. Then, we design the TSD network to predict their deviation to deform the initialized shape of the 3D grid, guided by the text prompt input.

**Network Design.** As shown in Fig. 3 (a), the inputs of TSD are text prompts and anchor positions. Firstly, the text prompts are encoded as the text embedding by an off-the-shelf text encoder, *e.g.*, CLIP (Radford et al., 2021) or T5 (Raffel et al., 2020). Considering the possibility of the complex input sentence, it remains non-trivial how to bridge each position and word in the sentence. The cross-attention (Vaswani et al., 2017) can quantify the correlation degree between each point and each word within a sentence. We then employ the cross-attention to design a module to obtain the deviation from the anchor position. It consists of the Layer Normalization (Ba et al., 2016), Multi-Head Attention, Feed-Forward Network, and shortcut connection (Vaswani et al., 2017). Consequently, certain positions, correlating more closely with corresponding words in the sentence, are assigned with higher attention scores. This process enables the aggregation of features that more accurately reflect the characteristics of the corresponding words. The detailed computation process is formulated as follows:

$$output = FFN(softmax(\frac{W_Q(p)W_K(y)}{\sqrt{d}}) \cdot W_V(y)), \tag{1}$$

where $p \in \mathbb{R}^3$ is the 3D coordinate of the anchor position, $y \in \mathbb{R}^{77 \times 1024}$ is the text embedding of the input prompt, 77 and 1024 are the sentence length and the embedding dimension, $W_Q(\cdot)$, $W_K(\cdot)$ and $W_V(\cdot)$ are the *query*, *key*, *value* transformation function, $d$ is the feature dimension, $FFN(\cdot)$ is the feed-forward network. The output of the TSD network is the offset $\Delta \in \mathbb{R}^3$ of the anchor positions. To ensure the stability of the training, we control the maximum extent to which a point can deviate from the anchor position. Specifically, given the degree of freedom $\beta \in \mathbb{R}$, we use the following equation to adjust the range of *output* into interval $(-\beta, \beta)$:

$$\Delta = 2 \cdot \beta \cdot sigmoid(output) - \beta. \tag{2}$$

Finally, the deformed position $p' \in \mathbb{R}^3$, which represents the centers of 3D Gaussian corresponding to input prompt, is formulated as follows:

$$p' = p + \Delta. \tag{3}$$

## 3.2 Text-guided Triplane Generator (TTG)

After determining the centers of the 3D Gaussians, we need to obtain the other four attributes. To efficiently assign features to each Gaussian, **the objective of TTG** is to **generate an implicit spatial representation in space**, represented by the triplane. Thus, we design a novel and highly efficient triplane generator that takes text prompts as input.

**One challenge** is that the previous triplane generation approaches, such as EG3D (Chan et al., 2022) and Instant3D (Li et al., 2024c), exhibit the problem of spatial inhomogeneity, as shown in Fig. 4. Since they directly segment a feature map into three feature maps along the channel dimension, only a few areas are computed together. For example, the position $(0, 0)$ in the 2D space is unfolded to $(0, 0, :)$, $(0, :, 0)$, and $(:, 0, 0)$, denoted by **blue** color in Fig. 4. Taking $1 \times 1$ *Conv* as an example, only these three areas are calculated together. On the contrary, $(0, 0, :)$ is hardly possible to be calculated with $(0, :, 1)$, because they do not appear at the same pixel in the 2D feature map. The same applies to the $3 \times 3$ *Conv*. This means that only a few areas share the same spatial information, while the others do not, thus causing spatial inhomogeneity. *For this, a simple yet effective way is to apply three generators (without sharing weights).*

**Another challenge** is that given the complex prompts, squeezing a sentence into a single style feature vector to apply AdaIN (Karras et al., 2019; 2020; 2021) could result in a loss of local details. Therefore, we need a more fine-grained generation method guided by the word level, thus can retain more information of text encoder trained on large-scale dataset. *Naturally, calculating cross-attention between the pixels of the feature map and words in the sentence is a better choice.*

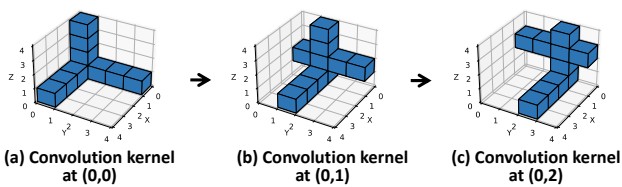

**(a) Convolution kernel at (0,0)**    **(b) Convolution kernel at (0,1)**    **(c) Convolution kernel at (0,2)**

Figure 4: The visualization of expanding 2D $1 \times 1$ convolution kernel (blue area) to 3D and its moving process in previous convolutional triplane generator Chan et al. (2022).

To address these two challenges, we design the Text-guided Triplane Generator (TTG), as shown in Fig. 3 (b). Our TTG is designed with the inspiration from the spatial transformer block and residual convolutional block in Stable Diffusion (Rombach et al., 2022). Considering the increased computational demand associated with pixel-wise self-attention in the feature map of (Rombach et al., 2022), we do not incorporate this layer into our network. Instead, we find that interleaved convolutional layers can sufficiently facilitate the interaction within the feature map. The detailed designs are shown in Fig. 5. For the whole pipeline, we first initialize a 2D input query according to its 2D trigonometric function position encoding. Three ResConv Blocks and Spatial Transformer blocks are stacked to assemble the prompt word features at low resolution through cross-attention. We then gradually increase the resolution of the feature map through the stacks of ResConv Blocks, Spatial Transformer blocks, and Upsample Block by five times, as depicted in Fig. 3 (b). Finally, we use a *Conv* layer to output the plane feature. We describe the design of the Spatial Transformer block, ResConv block, and Upsample block in detail.

**Spatial Transformer Block.** As Fig. 5 (a) shows, the Spatial Transformer Block comprises two multi-head cross-attention modules and a feed-forward network (Vaswani et al., 2017). The process is initiated

by flattening the 2D feature map into a 1D structure, thereby transforming the dimensions from $(H, W, C)$ to $(H \times W, C)$, with each pixel's feature considered as the input query embedding. Subsequent to this transformation, the features undergo normalization via Layer Normalization (Ba et al., 2016). The normalized features serve as queries, while the text embeddings act as keys and values in the computation of the cross-attention feature. This cross-modality attention mechanism is designed to align the feature map with the corresponding words in the input sentence. Following the application of two cross-attention modules, the features are further refined through a feed-forward network. This sequence of operations also incorporates the use of skip connections, mirroring the original transformer architecture (Vaswani et al., 2017), to facilitate effective feature processing and integration.

**ResConv Block.** As shown in Fig. 5 (b), the Residual Convolutional Block includes a Layer Normalization layer, a SiLU activation function (Elfwing et al., 2018), and $3 \times 3$ convolutional layers, with the skip connection between input and output.

**Upsample Block.** As depicted in Fig. 5 (c), the Upsample Block begins with interpolating the feature map to enlarge it by a factor of $2\times$, followed by processing through a $3 \times 3$ convolutional layer.

### 3.3 3D Gaussians Decoder

We aim to obtain four additional attributes of 3D Gaussian necessary for generation. Once the triplane, consisting of the three feature planes $\pi_{xy}, \pi_{xz}, \pi_{yz}$, is generated, we can obtain the feature vector $\mathcal{F} \in \mathbb{R}^{32}$ of each Gaussian based on its center $p'$. This feature vector is then converted into the additional attributes, including *opacity $\alpha \in \mathbb{R}$, scaling $S \in \mathbb{R}^3$, rotation $R \in \mathbb{R}^4$, and 1-order SH coefficient.*

Specifically, we first project the 3D coordinate onto three planes, X-Y, X-Z, and Y-Z. Based on the projected 2D coordinates, we can derive the features $\mathcal{F}_{xy}, \mathcal{F}_{xz}, \mathcal{F}_{yz}$ according to the interpolation with their four vertex in the 2D feature maps. To ensure the gradient back-propagation is evenly distributed across all three planes, we utilize an averaging operation to aggregate these fea-

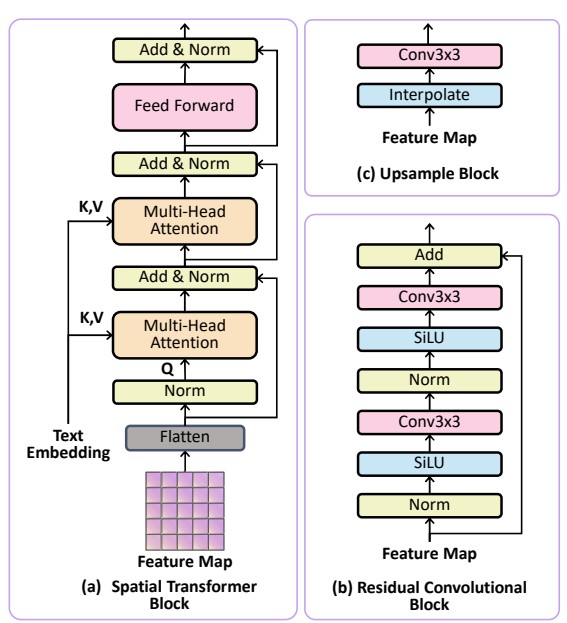

Figure 5: **A detailed illustration of specific blocks.** (a) Spatial Transformer Block. (b) Residual Convolutional Block. (c) Upsample Block.

tures, thereby obtaining the 3D Gaussian's feature $\mathcal{F}$. Given that the attributes of 3D Gaussian can be categorized into two groups, *i.e.*, shape and color, we develop two distinct transformation modules $F_{shape}$ and $F_{color}$. Each module is a lightweight, two-layer Multi-Layer Perceptron (MLP) network. To enhance the gradient back-propagation to the TSD, the center of 3D Gaussian $p'$ is additionally inputted into both modules:

$$\alpha, S, R = F_{shape}(\mathcal{F}, p'), \ SH = F_{color}(\mathcal{F}, p'). \tag{4}$$

In the training process, we observe the scaling $S$ is not a stable variation, and the memory consumption of 3D Gaussian rendering is extremely sensitive to it. Therefore, we use the following equation to control it to interval $(a, b)$:

$$S = (b - a) \cdot sigmoid(S) + a. \tag{5}$$

Upon obtaining all attributes of 3D Gaussian, our generation process is completed. We can render it from arbitrary view direction to 2D images.

### 3.4 Optimization

Our training commences with the selection of $B$ prompts from the training set. These prompts are then fed into our 3D Gaussians generator, which is tasked with generating the corresponding 3D GS representation of the objects. Following this, we proceed to randomly sample $C$ view directions to render $C$ 2D images. The $B \times C$ rendered images are supervised through the Score Distillation Sampling (SDS) loss function (Poole et al., 2022), as Eq. 6 shows, in conjunction with the Perp-Neg (Armandpour et al., 2023). In this way, our generator can gradually construct a mapping relationship between text and 3D.

$$\nabla_\theta \mathcal{L}(\phi, \mathbf{x} = g_\theta(y)) \triangleq \mathbb{E}_{t,\epsilon} \left[ w(t) \left( \hat{\epsilon}_\phi \left( \mathbf{z}_t; y', t \right) - \epsilon \right) \frac{\partial \mathbf{x}}{\partial \theta} \right], \tag{6}$$

where $y$ is the input prompt embedding of the generator, $\theta$ denotes the trainable parameters of 3D Gaussians generator, $\phi$ denotes the parameters of denoising network, $x$ is the generated image by process $g_\theta(\cdot)$, $w(t)$ is the weighting function with time step $t$ in the denoising schedule, $\epsilon$ is the random noise, $z_t$ is the noisy latents encoded from $x$, $y$ is the input prompt, and $y'$ adjusted text according to the sampling view direction, $\hat{\epsilon}_\phi(\cdot)$ is the predicted noise.

## 4 Experiments

This section first outlines the implementation details, inference speed, parameter count, and training dataset. The following **comparison experiments focus on four aspects**: **(I)** comparison with per-prompt text-to-3DGS, **(II)** fine-tuning the generated 3DGS model and then comparing with per-prompt text-to-3DGS, **(III)** comparison with amortized text-to-NeRF, **(IV)** comparison with text-to-image-to-3DGS methods. Finally, we present interesting findings regarding unseen words and conduct ablation studies to validate the network design. ***It's worth noting that all prompts in the experiments don't appear in the training dataset.***

### 4.1 Implementation Details

Our codebase is constructed on the PyTorch framework (Paszke et al., 2019) with Automatic Mixed Precision Training (AMP) and Gradient Checkpointing technology (Chen et al., 2016). We use the Adam optimizer (Kingma & Ba, 2014) with a constant learning rate of $5 \times 10^{-5}$, $\beta_1$ of 0.9 and $\beta_2$ of 0.99. We train our generator using the DeepFloyd

Table 1: The generation latency (millisecond) and rendering speed (FPS, Frames Per Second).

| Device | Generation Latency | Rendering Speed |
|---|---|---|
| RTX 3090 24GB | 79 ms | 698 FPS |
| A800 80GB | 77 ms | 705 FPS |

IF (Stablility, 2023) UNet to calculate the SDS Loss (Eq. 6). The prompt batch size is set to 64 and the camera batch size is set to 4. All our experiments are conducted on a server with 8 GPUs with 80GB memory. We set the freedom $\beta$ (Eq. 2) to 0.2, and the range of scaling $(a, b)$ (Eq. 5) to $(-9, -3)$. The anchor position is placed as a $64^3$ 3D grid, and the resolution of the generated triplane is $256 \times 256$. The generator is trained on a single prompt set including vehicle, daily life, and animal descriptions, in a total of 30K sentences. In Tab. 1, we show the inference latency on a single A800 GPU and a single RTX3090 GPU, which shows a large margin improvement, compared to optimization-based methods, which need **several hours**. Our generator contains about 500M trainable parameters. **We provide the training and inference pseudo code in *supplementary materials*.**

### 4.2 Comparison with other methods

We compare our method with previous text-to-3DGS methods. The **comparison experiments setting** includes four aspects: **(I)** comparison with per-prompt text-to-3DGS, including DreamGaussian (Tang et al., 2023), GSGEN (Chen et al., 2023b) and LucidDreamer (Liang et al., 2024) **(II)** fine-tuning the generated 3DGS model, and then compare with per-prompt text-to-3DGS, **(III)** comparison with amortized text-to-NeRF, including Instant3D (Li et al., 2024c), **(IV)** comparison with text-to-image-to-3DGS methods,

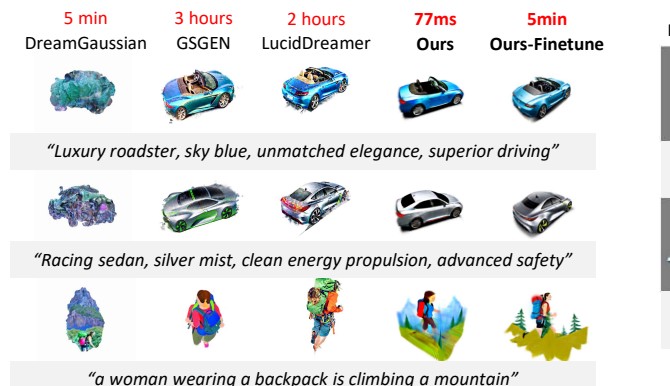

Figure 6: Comparison with per-prompt methods.

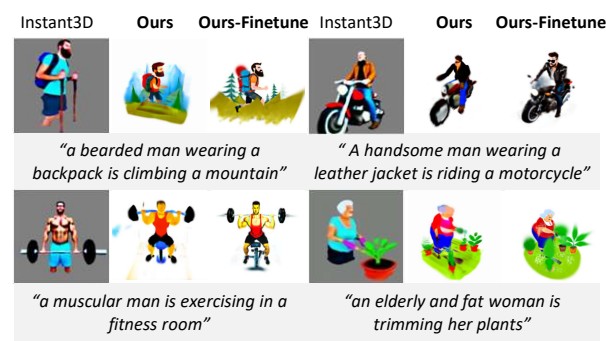

Figure 7: Comparison with amortized text-to-NeRF method.

including LGM (Tang et al., 2025) and GRM (Xu et al., 2024d). Besides, we present the quantitative results of human preference.

### 4.2.1 Per-prompt Text-to-3DGS

We select the current SoTA open-source text-to-3DGS methods, including DreamGaussian (Tang et al., 2023), GSGEN (Chen et al., 2023b) and LucidDreamer (Liang et al., 2024) for comparison. In this setting, we directly input the text prompt to BrightDreamer and use the same prompt to optimize the 3D GS model in other methods. As shown in the first four columns of Fig. 6, the other methods are hard to generate semantic-aligned 3D GS models from complex text prompts, primarily due to their limited capability in comprehending intricate text inputs. A similar conclusion can be found in the works (Jiang et al., 2024; Zhou et al.; Cheng et al.; Bai et al., 2023; Epstein et al.). It is hard to optimize a 3D GS model for a prompt containing multiple objects. Our method, trained across a diverse distribution of multiple text prompts, exhibits a robust capability to comprehend complex prompts effectively.

### 4.2.2 Quickly Finetuning after generation

We demonstrate that the 3D GS model generated by BrightDreamer can be efficiently fine-tuned using per-prompt optimization methods to enhance quality. Specifically, we use the ISM (Liang et al., 2024) loss to optimize the generated 3D GS model for several hundred iterations. As evidenced in the last column of Fig. 6, the third and sixth columns of Fig. 7, the model's quality of details significantly improves within just **several minutes**.

### 4.2.3 Amortized Text-to-NeRF

We compare BrightDreamer with SoTA open-source amortized Text-to-NeRF methods, Instant3D (Li et al., 2024c). It is worth noting that our method is the first amortized Text-to-3DGS method. Moreover, our method inherits numerous advantages of 3D GS over NeRF. Since ATT3D (Lorraine et al., 2023) and Latte3D (Xie et al., 2024) don't open their code and demo, we don't compare with them. As

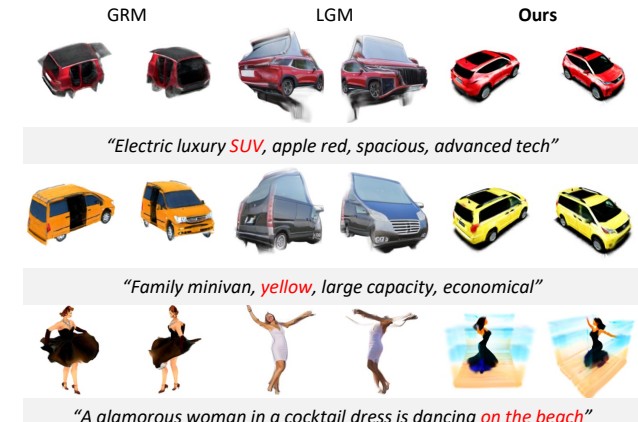

Figure 8: Comparison with text-to-image-to-3DGS methods.

shown in Fig. 7, our semantic alignment is better. Furthermore, benefiting from the training efficiency of 3D GS, we can quickly fine-tune the generated 3D GS model to enhance the details.

### 4.2.4   Text-to-image-to-3DGS

Besides the direct generation of text-to-3DGS, alternative methods also exist to achieve this goal. For instance, LGM (Tang et al., 2025) and GRM (Xu et al., 2024d) first generate the reference images using a text-to-image diffusion model, followed by the creation of a 3D model from these images. However, since the second phase of image-to-3DGS is trained on the limited 3D dataset, it is hard for them to generate the 3D GS models for complex prompt input. As shown in Fig. 8, we demonstrate the comparison result. Furthermore, since these methods require an intermediate image generation step, they are considerably slower than our BrightDreamer approach.

### 4.2.5   Quantitative Comparison

As shown in Tab. 2, we provide the percentage numerical comparison of human preference choice. We present participants with five rendered videos along with the corresponding text prompts generated by five baseline models, allowing them to select their preferred option for each case. Most of the users expressed a preference for the content generated by our model. Moreover, we also render the 10-view images to compute CLIP similarity as the CLIP score.

| Method | Instant3D | LucidDreamer | **Ours** |
|---|---|---|---|
| Preference | 37.3% | 7.2% | **55.4%** |
| CLIP Score | 0.322 | 0.286 | **0.324** |

Table 2: Human Preference.

### 4.3   Discussion about Generalizability

Beyond sentence-level generalizability, our findings indicate that BrightDreamer can also correctly interpret **some unseen words**. As demonstrated in Fig. 9, although the word "banana" is absent from our training dataset, BrightDreamer successfully identifies its color. This capability not only suggests a potential for broad word-level generalizability but also confirms that BrightDreamer effectively learns the mapping from text distribution to 3D GS distributions.

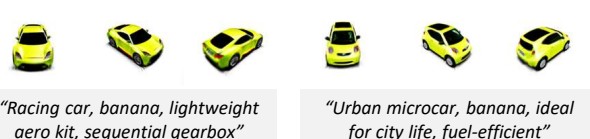

"Racing car, banana, lightweight aero kit, sequential gearbox"          "Urban microcar, banana, ideal for city life, fuel-efficient"

Figure 9: Demonstration of word-level generalizability. The word "banana" **never appears** in the training set.

### 4.4   Ablation Studies

As shown in Fig. 10, we validate our network design for training. Compared between Fig. 10 (a) and Fig. 10 (b), our divided triplane generator can reduce the degree of chaos in the space significantly, which shows the necessity of our division. Notably, we adjust the number of blocks, *i.e.*, network depth, to achieve a similar number of parameters. As Fig. 10 (a) and Fig. 10 (c) demonstrate, it is necessary to pass the coordinate into the $F_{shape}$ and $F_{color}$. This design can construct a gradient pathway toward the TSD Network, ensuring more accurate shape formulation.

## 5   Conclusion

In this paper, we introduce the first text-driven 3D Gaussians generative framework, BrightDreamer, capable of generating 3D Gaussians within a remarkably low latency of 77ms. To efficiently generate millions of 3D Gaussians, we innovatively deform anchor positions and use these new positions as centers based on the input prompt. This approach successfully overcomes the challenge of generating a large number of positions. Regarding network architecture, we thoroughly reevaluate the triplane generation process and introduce an improved alternative strategy. Our key contribution is to greatly enhance the generalized 3D generation, offering a novel and efficient way to create 3D assets from text prompts instantly. Extensive experiments demonstrate that BrightDreamer possesses strong semantic understanding and generalization abilities. Due to the limited 3D data, developing methods based on 2D diffusion is an important direction.

**Future work.** The spatial resolution of the generated 3D model is relatively low, resulting in a lack of fine-grained detail. To address this problem, we will focus on reducing the gpu memory occupation of SDS loss to achieve higher spatial resolution. Besides, since the LGM (Tang et al., 2025) and GRM (Xu et al., 2024d) can generate highly detailed single-object 3D GS models while our method can generate semantic-rich 3D GS models, developing a method that combines these two advantages is an interesting research direction. We will also collect broader data to train a larger model. Furthermore, we will also incorporate the better SDS loss design into our framework to achieve higher quality.

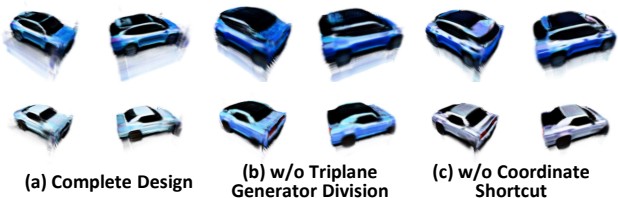

(a) Complete Design    (b) w/o Triplane Generator Division    (c) w/o Coordinate Shortcut

Figure 10: **Ablation studies.** All models are trained for 10,000 iterations (incomplete training) with the same configuration: (a) Our complete design, (b) Single generator replacing three separate generators, (c) Coordinate input removed from $F_{shape}$ and $F_{color}$.

## Acknowledgments

This work is supported by the MOE AcRF Tier 1 (Call 2/2025) Grant under Grant No. RG160/25.

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

## A    Training Details of BrightDreamer

For a clearer illustration of our training pipeline, we provide the details in Algorithm. 1. We train our BrightDreamer for about 2 days on a server with 8 GPUs with 80GB memory. Actually, a lower batch size and less training time can also result in similar quality.

---

**Algorithm 1** Training Procedures of BrightDreamer

---

**Input**: $\mathbf{S}$, training prompts set ; $B$, batch size of prompts; $C$, batch size of cameras; $max\_iter$, maximum training iterations

**for** $i \leftarrow 1$ to max_iter **do**

  $prompts \leftarrow$ sample B prompts in $\mathbf{S}$;

  $3D\_GS \leftarrow$ BrightDreamer($prompts$);

  $loss \leftarrow 0$;

  **for** $i \leftarrow 1$ to B **do**

    $cameras \leftarrow$ randomly sample $C$ cameras;

    $images \leftarrow$ render($3D\_GS[i]$, $cameras$);

    $texts\_dir \leftarrow prompt[i]$ added front/side/back;

    $loss \leftarrow loss +$ SDS($images$, $texts\_dir$);  # Eq. 6

  **end for**

  optimizer.zero_grad();

  $loss$.backward();

  optimizer.step();

**end for**

**End**.

---

## B    Inference Procedures of Our BrightDreamer

In Algorithm. 2, we provide the inference details of our BrightDreamer.

## C    Discussion about Generalization

In Fig. 2 of the main paper, we show the different combinations that don't appear in the training prompts, *e.g.*, 'deep purple' and 'light purple'. Here, we show the word that doesn't appear in the training prompts may also be understood. For example, 'banana' doesn't appear in the training process anymore. However, as shown in Fig. 19 and Fig. 27, it can generate the corresponding color accurately.

## D    More Visual Results

From Fig. 12 to Fig. 51, we provide more multi-view results of our BrightDreamer. Our method shows strong detail control ability.

---
**Algorithm 2** Inference Procedures of Our BrightDreamer

---
**Input**: $p$, the anchor positions; $prompt$, input text prompt;

**Output**: $3D\_GS$, the generated 3D GS;

# Shape Deformation

$\Delta \leftarrow$ TSD($p$, $prompt$);

$p' \leftarrow p + \delta$;

# Triplane Generation
$\pi_{xy} \leftarrow$ TTG_XY($prompts$);

$\pi_{xz} \leftarrow$ TTG_XZ($prompts$);

$\pi_{yz} \leftarrow$ TTG_YZ($prompts$);

# 3D Gaussian Decoding

$\mathcal{F}_{xy} \leftarrow$ grid_sample($p'[..., [0, 1]]$, $\pi_{xy}$);

$\mathcal{F}_{xz} \leftarrow$ grid_sample($p'[..., [0, 2]]$, $\pi_{xz}$);

$\mathcal{F}_{yz} \leftarrow$ grid_sample($p'[..., [1, 2]]$, $\pi_{yz}$);

$\mathcal{F} \leftarrow (\mathcal{F}_{xy} + \mathcal{F}_{xy} + \mathcal{F}_{xy})$ / 3;

$S$, $R$, $\alpha \leftarrow F_{shape}(\mathcal{F}, p')$;

$SH \leftarrow F_{color}(\mathcal{F}, p')$;

# 3D GS Construction

$3D\_GS \leftarrow$ GaussianModel();

$3D\_GS.\_xyz \leftarrow p'$;

$3D\_GS.\_opacity \leftarrow \alpha$;

$3D\_GS.\_rotation \leftarrow S$;

$3D\_GS.\_scaling \leftarrow R$;

$3D\_GS.\_feature\_dc \leftarrow SH$;

**return** $3D\_GS$;

**End**.

---

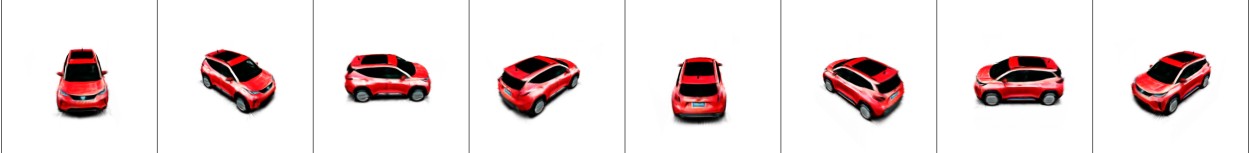

Figure 11: Electric luxury SUV, apple red, spacious, advanced tech

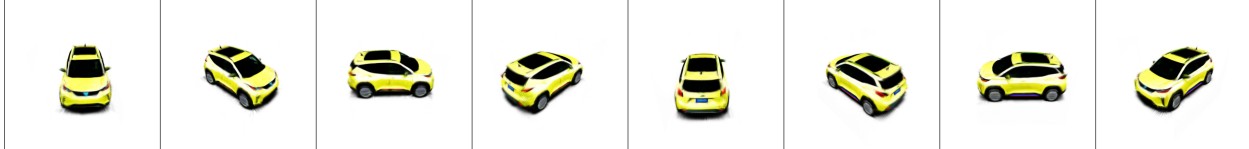

Figure 12: Electric luxury SUV, yellow, spacious, advanced tech

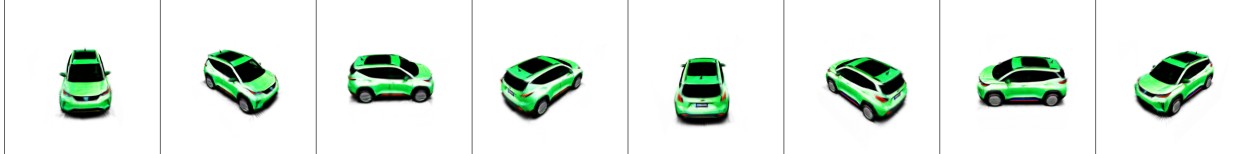

Figure 13: Electric luxury SUV, forest green, spacious, advanced tech

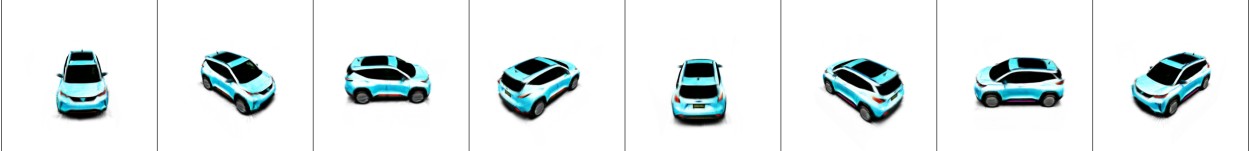

Figure 14: Electric luxury SUV, cyan, spacious, advanced tech

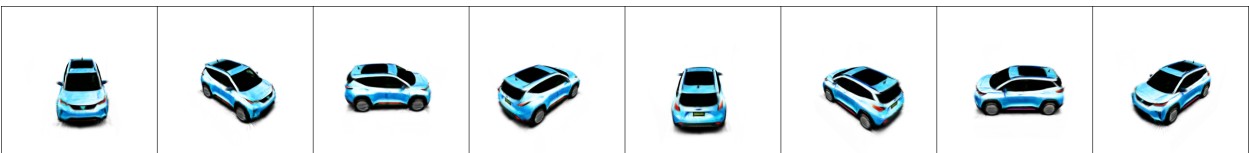

Figure 15: Electric luxury SUV, deep blue, spacious, advanced tech

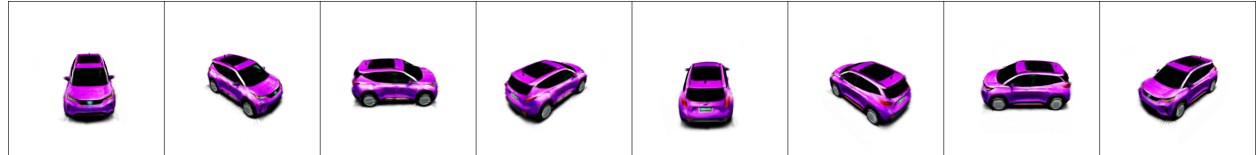

Figure 16: Electric luxury SUV, light purple, spacious, advanced tech

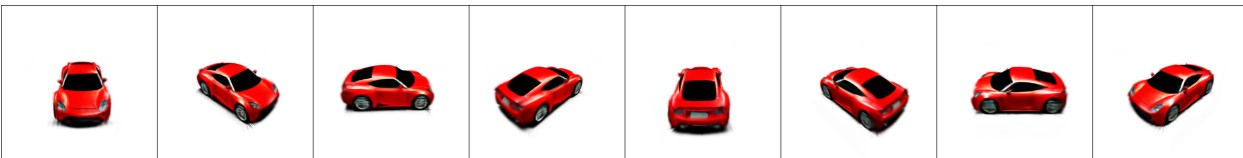

Figure 17: Racing car, deep red, lightweight aero kit, sequential gearbox

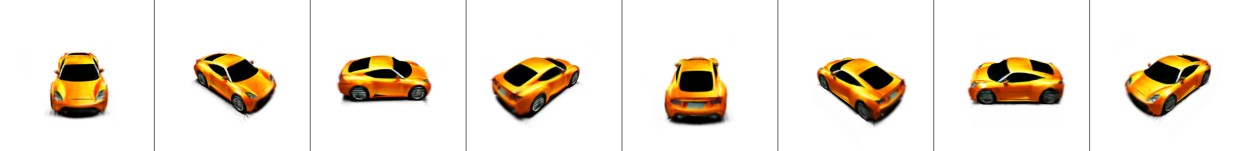

Figure 18: Racing car, blaze orange, lightweight aero kit, sequential gearbox

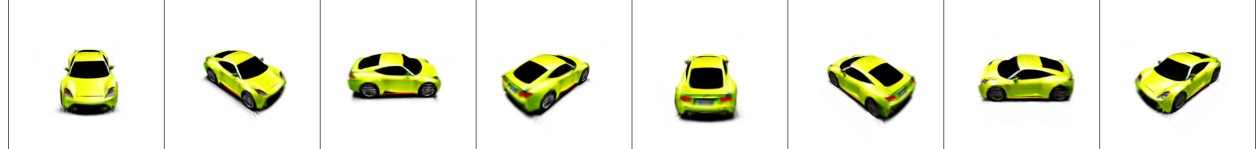

Figure 19: Racing car, banana, lightweight aero kit, sequential gearbox

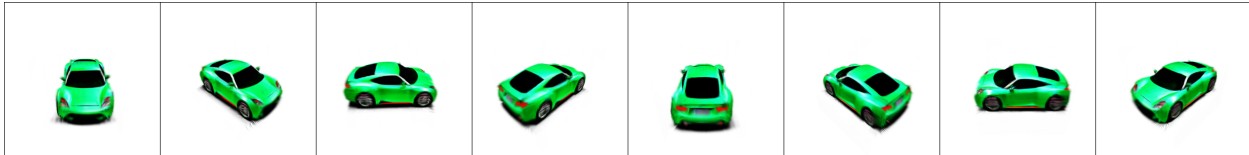

Figure 20: Racing car, green, lightweight aero kit, sequential gearbox

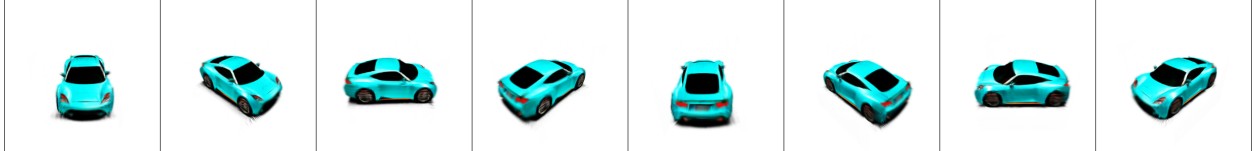

Figure 21: Racing car, cyan, lightweight aero kit, sequential gearbox

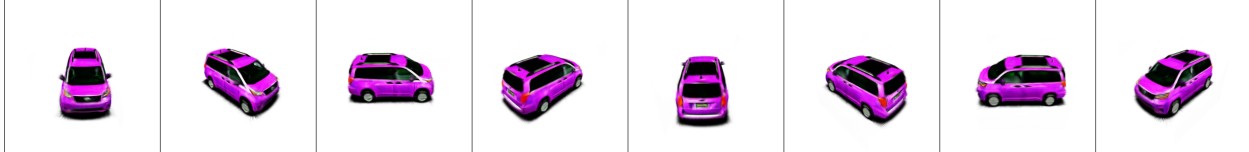

Figure 22: Family minivan, purple, large capacity, economical

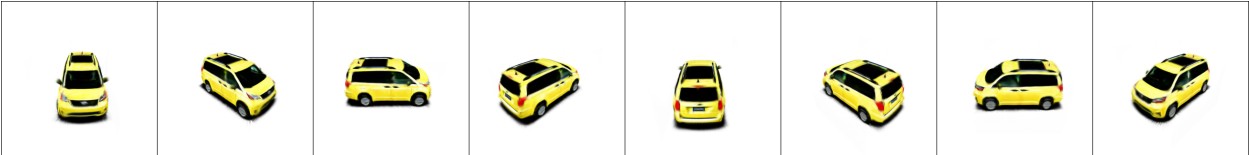

Figure 23: Family minivan, yellow, large capacity, economical

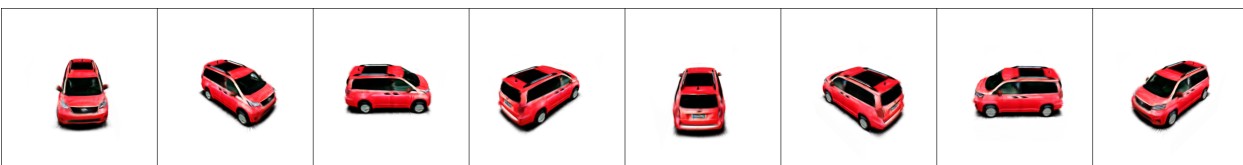

Figure 24: Family minivan, apple red, large capacity, economical

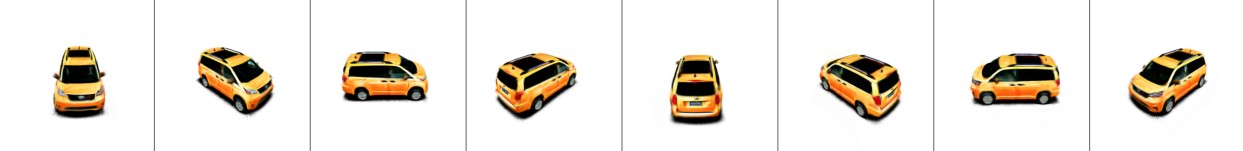

Figure 25: Family minivan, orange, large capacity, economical

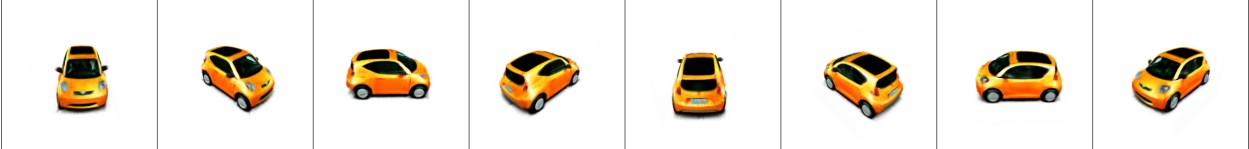

Figure 26: Urban microcar, orange, ideal for city life, fuel-efficient

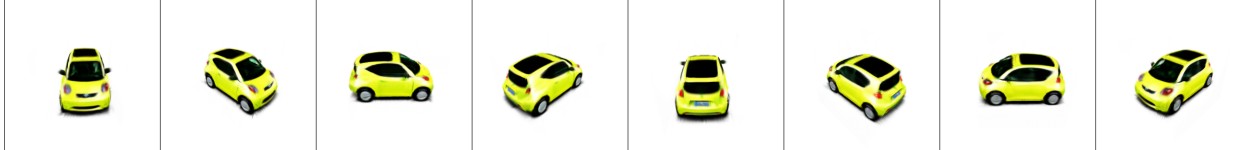

Figure 27: Urban microcar, banana, ideal for city life, fuel-efficient

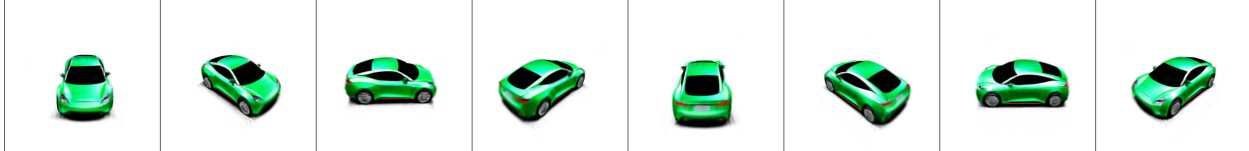

Figure 28: Electric coupe, forest green, sleek design, autonomous features

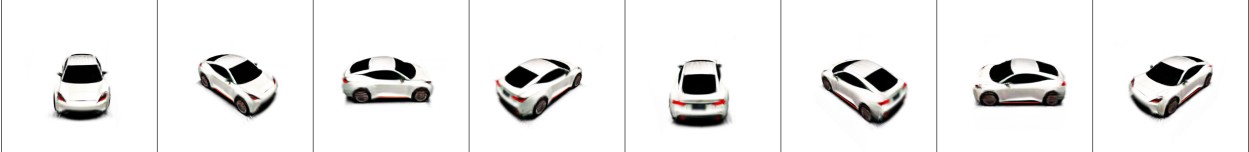

Figure 29: Electric coupe, pearl white, sleek design, autonomous features

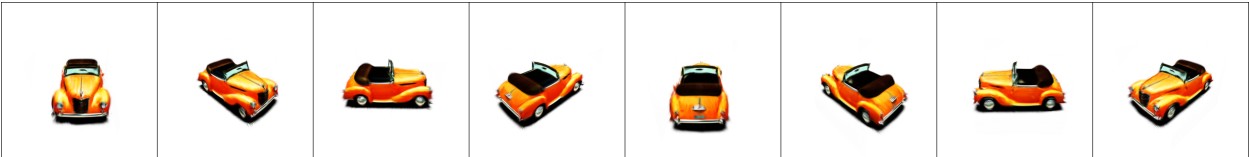

Figure 30: Vintage convertible, orange, chrome bumpers, white-wall tires

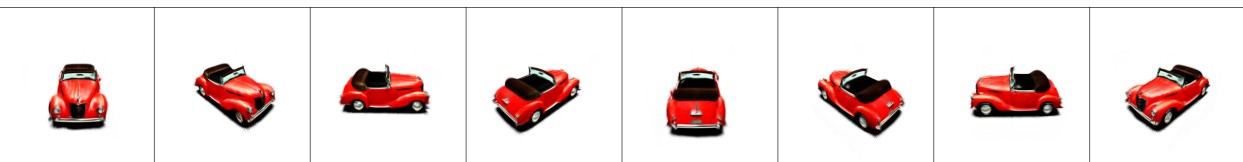

Figure 31: Vintage convertible, apple red, chrome bumpers, white-wall tires

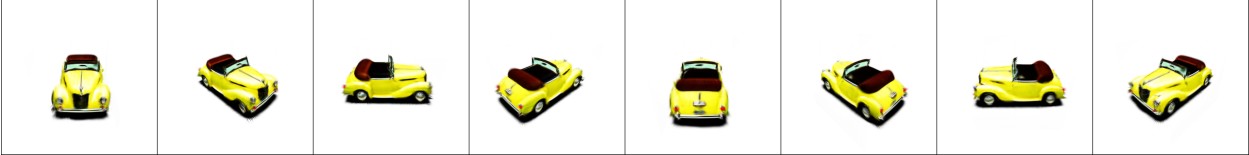

Figure 32: Vintage convertible, yellow, chrome bumpers, white-wall tires

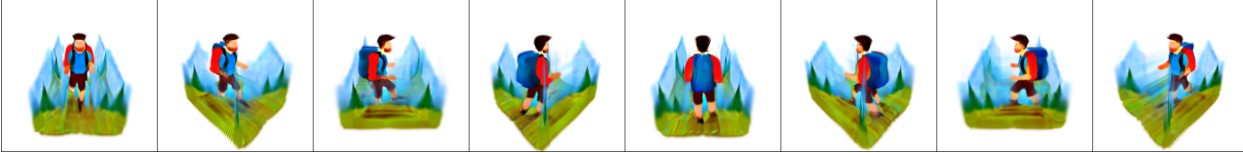

Figure 33: a man wearing a backpack is climbing a mountain

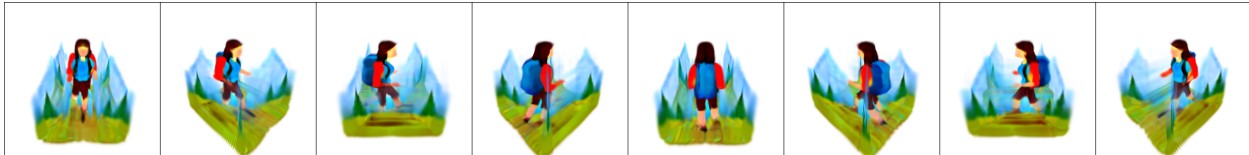

Figure 34: a woman wearing a backpack is climbing a mountain

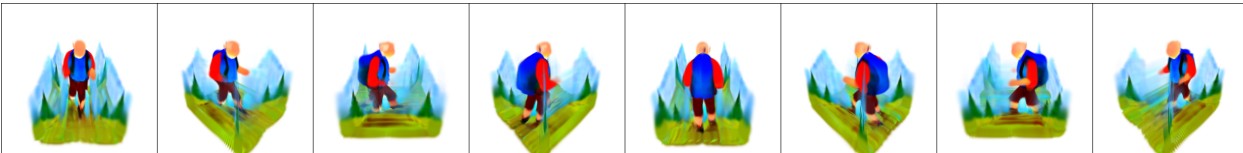

Figure 35: an elderly man wearing a backpack is climbing a mountain

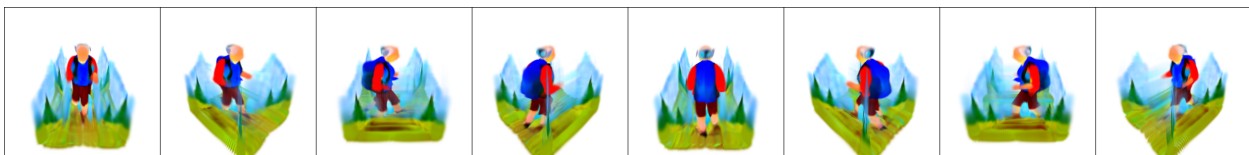

Figure 36: an elderly woman wearing a backpack is climbing a mountain

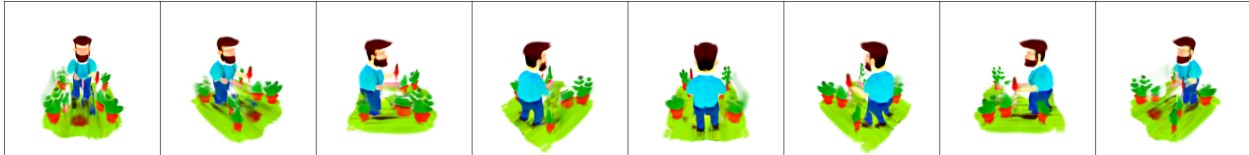

Figure 37: a man is trimming his plants

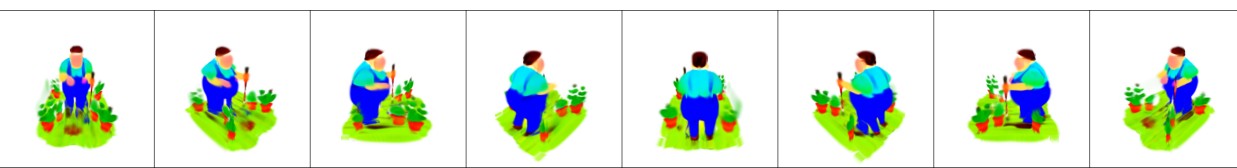

Figure 38: a fat man is trimming his plants

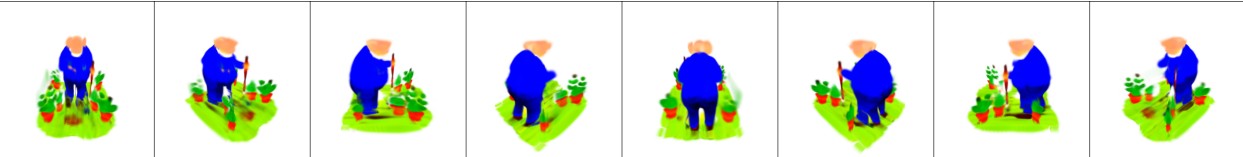

Figure 39: a fat and elderly man is trimming his plants

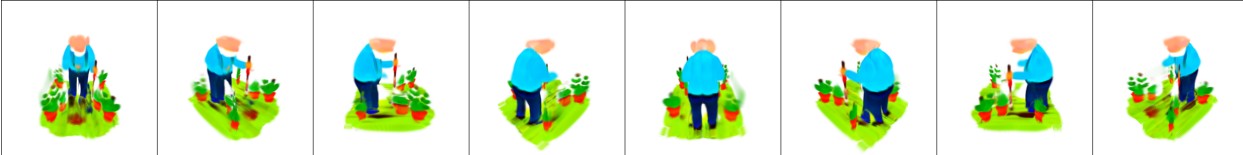

Figure 40: an elderly man is trimming his plants

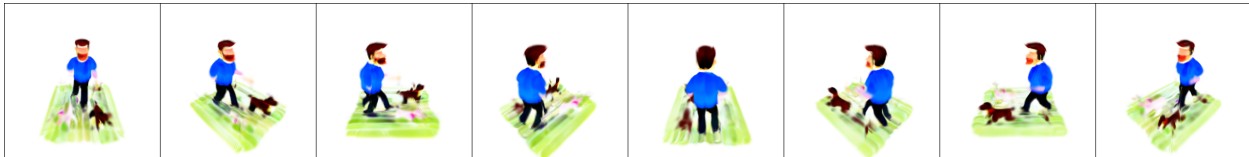

Figure 41: a man is playing with a dog

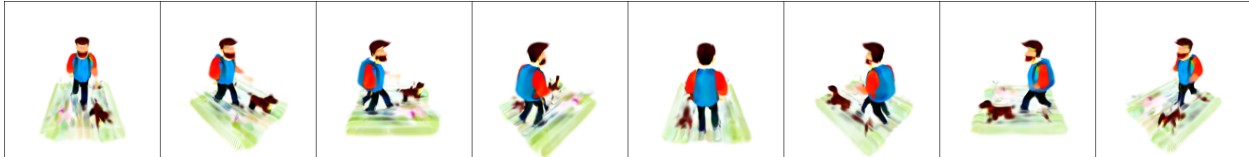

Figure 42: a man wearing a backpack is playing with a dog

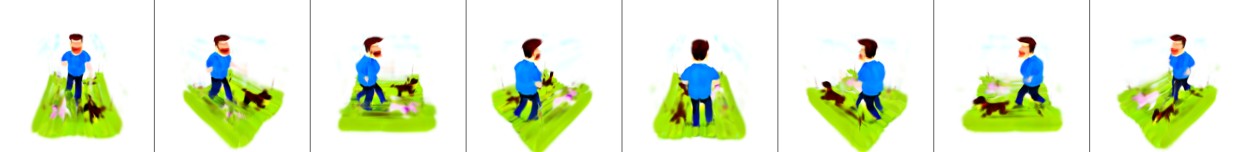

Figure 43: a man is playing with a dog on the lawn

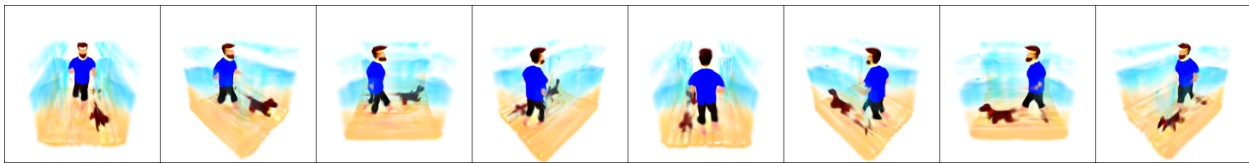

Figure 44: a man is playing with a dog on the beach

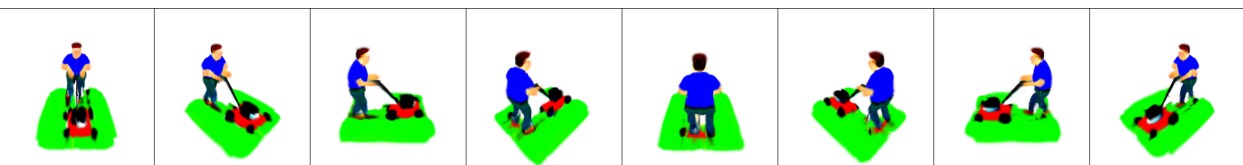

Figure 45: a man is mowing the lawn

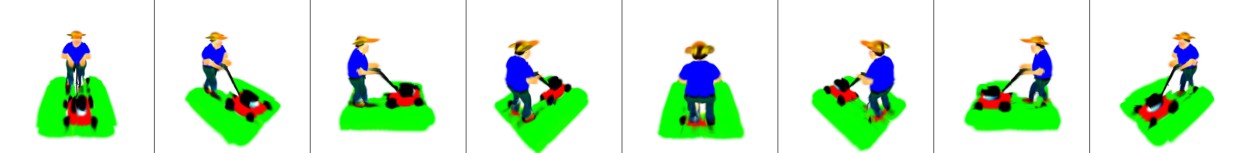

Figure 46: a man wearing a hat is mowing the lawn

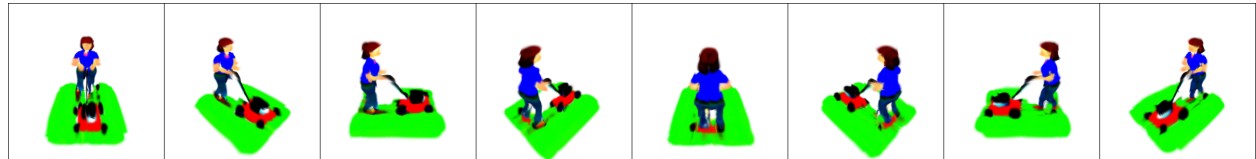

Figure 47: a woman is mowing the lawn

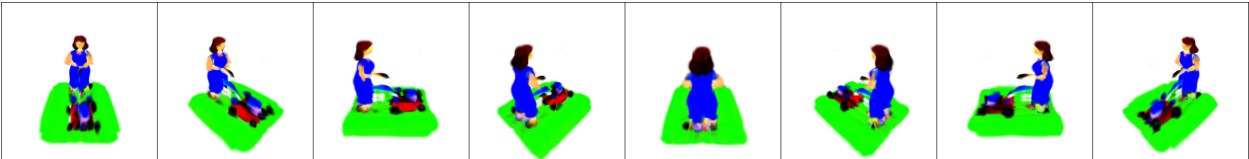

Figure 48: a woman in a long dress is mowing the lawn

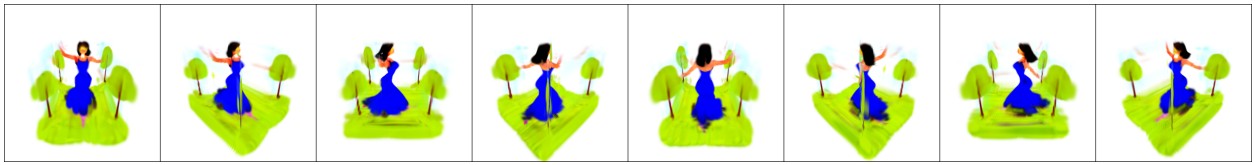

Figure 49: A glamorous woman in a cocktail dress is dancing in the park

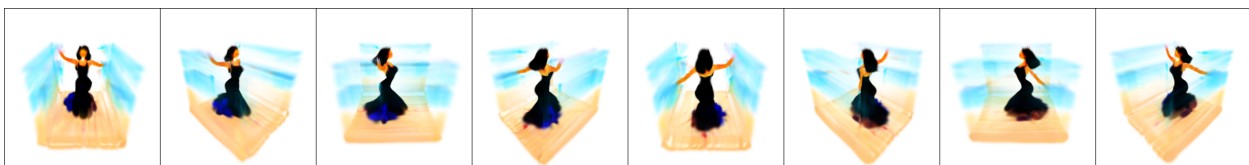

Figure 50: A glamorous woman in a cocktail dress is dancing on the beach

