# OpenReview forum: "BrightDreamer: Generic 3D Gaussian Generative Framework for Fast Text-to-3D Synthesis"
_TMLR — Accepted by TMLR_

### Review · Reviewer_CAG7 · 2025-11-19

**Summary Of Contributions:**

The paper proposes "BrightDreamer," a feed-forward generative framework designed to synthesize 3D Gaussian Splatting (3DGS) representations directly from text prompts. Unlike optimization-based methods (e.g., DreamFusion, DreamGaussian) that require time-consuming per-prompt optimization, BrightDreamer aims to generate 3D assets in approximately 77ms after a one-time training phase.
The methodology consists of two primary components:
Text-guided Shape Deformation (TSD): A network that deforms a set of fixed anchor positions (a 3D grid) into the centers of 3D Gaussians based on the input text embedding.
Text-guided Triplane Generator (TTG): A modified triplane generator that addresses spatial inhomogeneity by using separate generators for each plane (XY, XZ, YZ) and integrates text embeddings via cross-attention.
The resulting triplanes are decoded into Gaussian attributes (opacity, scaling, rotation, SH). The model is trained using Score Distillation Sampling (SDS) and a Perp-Neg loss. The authors demonstrate that their method is significantly faster than per-prompt optimization methods and shows better semantic alignment for complex prompts than standard optimization baselines.

**Audience:**

Yes

**Audience Explanation:**

The intersection of text-to-3D generation and 3D Gaussian Splatting is currently a highly active area of research, making this work relevant to a broad section of the ML community. Researchers will be particularly interested in the proposed feed-forward architecture, which effectively addresses the significant latency bottlenecks found in prevailing optimization-based methods. Furthermore, the insights regarding direct parameter prediction for unstructured 3D representations offer valuable contributions to the field of generative modeling.

**Broader Impact Concerns:**

The paper presents a technical framework for accelerating text-to-3D generation. I do not see any specific ethical concerns or negative societal impacts arising from this work.

**Claims And Evidence:**

Yes

**Claims Explanation:**

The paper substantiates its efficiency claims with benchmarks showing sub-100ms generation times and high rendering frame rates (Table 1). Qualitative comparisons against state-of-the-art baselines convincingly demonstrate the model's superior ability to interpret complex, compositional text prompts without significant artifacts. Additionally, the ablation studies in Section 4.4 provide clear empirical support for the effectiveness of the proposed architectural contributions, such as the split triplane generator.

**Requested Changes:**

1. The evaluation is deficient in standard quantitative metrics, such as CLIP-Score or FID, and lacks testing on a large-scale held-out set. Relying solely on subjective user studies prevents an objective comparison of generation fidelity and text alignment against baselines.

2. The fixed 256x256 triplane resolution acts as a significant bottleneck, severely constraining the geometric complexity and texture sharpness of the outputs. This limitation prevents the framework from scaling effectively to generate high-fidelity 3D assets.

3. The assertion that using separate generators for each axis resolves "spatial inhomogeneity" lacks theoretical justification or variable isolation. The observed improvements in the ablation study may simply result from the increased parameter count rather than the proposed architectural division.

4. The training dataset appears limited to specific closed-set categories like vehicles, which severely restricts open-vocabulary generalization. Consequently, the model likely overfits to this narrow distribution and will struggle to generate concepts outside these specific domains.

5. The generated samples exhibit significant mode collapse, with objects within the same category sharing nearly identical geometries despite varying prompts. The model appears to retrieve a mean shape rather than learning a diverse generative distribution.

6. The claim that coarse outputs serve as effective initializations is contradicted by the visual results, which show negligible geometric improvement after refinement. Starting from such low-fidelity geometry traps the optimization in local minima, preventing the recovery of sharp structural details.

---

> ### Author Response · Authors · 2025-12-08
> **Response to Requested Changes**
>
> Sincerely, thanks for reviewing our paper and providing constructive suggestions to help us improve the paper. We have fixed the problem and made clearer statements according to your requested changes, highlighted in cyan in the revised paper. We hope our responses adequately address your concerns, and we remain available for any further discussion or clarification you may require.
>
> ## Q1 Quantitative Comparison Based on CLIP-Score
>
> Thanks for your suggestions. We render the multi-view images to compute the clip feature similarity with the input prompt. And the results are shown in the table below.
>
>
> | Method     | Instant3D | LucidDreamer |  Ours |
> |------------|:---------:|:------------:|:-----:|
> | CLIP Score |   0.322   |     0.286    | 0.324 |
>
>
> ## Q2 Resolution
>
> We have made significant efforts to achieve 256 resolution, including experimentally removing the self-attention module that consumes substantial GPU memory. The primary memory consumption stems from the subsequent SDS computation, which requires training an SD v2 model. Therefore, reducing the amount of SDS memory used is the only way to potentially increase the resolution further. We leave this as future work.
>
> ## Q3 Ablation Studies
>
> Sorry for the unclear illustration. We maintained an approximate consistency in the number of parameters by adjusting the number of blocks.
>
> ## Q4 Larger Dataset
>
> We are currently limited to generating within domains with similar datasets. However, Fig. 9 demonstrates that our model can still achieve a correct understanding of unseen vocabulary. This indicates that our model still has room for further expansion. And our scale of corpus has surpassed previous works such as ATT3D and Instant3D. To collect a larger corpus and train larger models, we leave it as future work.
>
> ## Q5 Mode Collapse
>
> We can generate different vehicles and people based on prompts, such as “SUV,” “Racing car,” “Family minivan,” “old man,” “a man wearing a backpack,” etc. And the generated content aligns with the semantic meaning of the input prompt.
>
> ## Q6 Local Minima
>
> It does not get stuck in local optima, since a text-aligned 3D model has already been generated. Methods like LucidDreamer cannot produce semantically coherent scenes even after hours of processing. Given our initialization, subsequent SDS optimization with a 3DGS splitting strategy significantly enhances spatial resolution and sharpens details.

---

### Review · Reviewer_6PBk · 2025-11-24

**Summary Of Contributions:**

### **Summary**

This paper explores a training-based, feed-forward alternative to existing optimization-driven text-to-3D approaches (e.g., SDS, VSD). Instead of optimizing a new model for every prompt, the proposed method directly generates 3D Gaussian splats from text, enabling both fast inference and applicability to unseen prompts. Qualitative comparisons against four representative types of text-to-3D baselines suggest improvements in both speed and output quality.

### **Strengths**

* The paper is clearly written and easy to follow. Each component is well-motivated, and notations and architectural choices are presented in a clear and structured way.
* The comparisons with related work are comprehensive and span different classes of text-to-3D methods, which makes the empirical claims more convincing.

### **Weaknesses**

* Although many results look visually appealing, several outputs appear biased toward simplified, cartoon-like styles. This seems related to an "over-optimization" issue where finer textures and colors collapse into clean, minimalistic shapes. While this improves text-3D alignment, it may restrict the method’s usefulness when users expect richer details.
* The comparisons cover mainly "car" and "human"-related prompts. Evaluating a broader, more diverse set of object categories would help validate generalizability. For instance, baselines such as LucidDreamer can handle avatars, portraits, and other domains; showing results across these types would strengthen the claim of broad applicability, especially given the qualitative nature of text-to-3D evaluation.

**Additional Comments:**

no

**Audience:**

Yes

**Audience Explanation:**

Text-to-3D generation is an active and rapidly evolving area of interest for both researchers and digital content creators. This paper introduces a feed-forward, optimization-free pipeline, which represents a meaningful shift from the dominant per-prompt optimization paradigm. Given the substantial reduction in generation time and the practical implications for interactive or large-scale applications, the findings are likely to attract attention from the TMLR community.

**Broader Impact Concerns:**

It would be helpful for the authors to include a brief statement confirming that the training data does not contain sensitive or personally identifiable information. It may also be worthwhile to clarify that the goal of the work is to advance research in efficient text-to-3D generation, and that the model is not intended for malicious or harmful applications.

**Claims And Evidence:**

No

**Claims Explanation:**

The paper claims improvements in both speed and quality. The evidence for the speed claim is clear and convincing, as demonstrated in Table 1, where the reported latency improvements are substantial. However, the evidence for the quality and generalizability claims is less convincing. Although Figures 6 to 8 show competitive or better results compared to baselines, many outputs appear over-optimized toward simplified, cartoon-like styles. In addition, the qualitative evaluations focus largely on cars and humans, without demonstrating broader category diversity. Without more varied and detailed examples, the claim of strong generalizability remains insufficiently supported.

**Requested Changes:**

### **Paper Content** (Critical)
* The paper should address the observed over-optimization issue more explicitly. In particular, it would help if the authors could analyze the underlying causes (e.g., training dynamics, loss formulation, dataset bias) and describe potential solutions or mitigations. Updated results reflecting such improvements would further strengthen the contribution.
* The qualitative evaluation would benefit from broader coverage across diverse object categories. Including comparisons beyond cars and humans would make it easier for readers to assess generalizability and better support the claims made in the paper.

### **Paper Format** (Minor)
* Some paragraphs following equations appear to have unnecessary indentation (e.g., after Equations 1 and 6). Revising these would improve readability.
* It is recommended to update the citation information to reflect the latest accepted venues. For instance, Instant3D is now accepted to ICLR 2024, and LucidDreamer is accepted to CVPR 2024. This helps readers understand the maturity of the referenced work and properly contextualizes the baselines.

---

> ### Author Response · Authors · 2025-12-08
> **Response to Requested Changes**
>
> Sincerely, thanks for reviewing our paper and providing constructive suggestions to help us improve the paper. We have fixed the problem and made clearer statements according to your requested changes, highlighted in **cyan** in the revised paper. We hope our responses adequately address your concerns, and we remain available for any further discussion or clarification you may require.
>
> ## Q1 Over-optimization Issue
>
> Actually, since we are based on vanilla SDS, encountering this issue is unavoidable. The original SDS also exhibits this problem when optimizing one by one (please refer to its original paper, Dreamfusion), and many variants face the same challenge. To completely avoid this issue, we would need to integrate subsequent photo-realistic SDS designs into our framework. We leave this as future work.
>
> ## Q2 Broader Data Coverage
>
> We trained our model using three types of mixed text data to demonstrate its generalization capabilities. This represents a significantly larger text corpus than previous comparable works (ATT3D, Instant3D). Training larger generative networks requires substantially more data and computational resources, which we identify as a future research direction.
>
> ## Q3 Format Problem
>
> Thanks for your suggestions. We have fixed the format problem.
>
> ## Q4 Citation Update
>
> Thanks for your suggestions. We have fixed this problem. There are two works named “Instant3D”. We have updated their citation information in IJCV and ICLR.

---

> > ### Comment · Reviewer_6PBk · 2026-01-17
> > **Response to Authors' Reply**
> >
> > Thanks to the authors for addressing the requested changes. Points 1, 3, and 4 are resolved. However, regarding point 2, I'm still not fully convinced by the response.
> >
> > The authors state that they use a larger dataset than prior work, but the paper currently presents results from a fairly limited set of categories. In comparison, earlier methods (e.g., Instant3D) demonstrate broader category coverage (e.g, *hippo* and *eggs* in Figure 1). Could the authors provide additional qualitative examples beyond the current category to better support this claim?
> >
> > This concern aligns with Reviewer CAG7’s comment, and addressing it would strengthen the third contribution that the proposed approach can handle complex scenarios.

---

> > > ### Author Response · Authors · 2026-01-17
> > >
> > > We apologize for any confusion caused by the naming collision. There are currently **three distinct papers** sharing the title "Instant3D":
> > >
> > > [1] Instant3D: Instant Text-to-3D Generation, IJCV
> > >
> > > [2] Instant3d: Fast text-to-3d with sparse-view generation and large reconstruction model, ICLR
> > >
> > > [3] Instant-3d: Instant neural radiance field training towards on-device ar/vr 3d reconstruction, ISCA
> > >
> > > We believe **the work you mentioned is the second one (ICLR)**. That method employs a text-to-multi-view-to-image pipeline, which demands substantial computational resources (utilizing **32 A100s** for the multi-view diffusion model and **128 A100s of 7 days** for the reconstruction model) and relies on large-scale **image-3D paired datasets**. It does not match the setting of our paper and cannot be used as our baseline
> > >
> > > In contrast, **our method focuses on distilling 2D diffusion models using only text data**, which significantly reduces GPU requirements and data costs. Given these distinct methodological differences, we adopted the first Instant3D (IJCV) as our primary baseline. Under similar settings, our method achieves superior results. Compared to the baseline, our network demonstrates stronger generalization capabilities. Furthermore, we have integrated 3D Gaussian Splatting (3DGS) into our pipeline to enable a faster and higher-quality 3D representation. Our method also demonstrates that general SDS training can be extended to multiple categories of objects, not just single-class objects, like ATT-3D and Instant3D (IJCV).
> > >
> > > We sincerely appreciate the time and effort you have dedicated to reviewing our paper. We welcome any further discussion and are happy to address additional questions.

---

> > > > ### Comment · Reviewer_6PBk · 2026-01-17
> > > > **Response to Authors' Reply**
> > > >
> > > > Thanks for the clarification. It makes sense to treat compute/resources as an important factor when discussing the baseline. That said, since the proposed method is distilled from a diffusion model, I would expect the resulting model to potentially generalize to a broader set of categories.
> > > >
> > > > Could the authors share at least a few simple results on additional categories, or briefly describe what happens when testing beyond the current scope? For example, are the outputs generally successful but lower in quality, or do they fail outright in other categories?

---

> > > > > ### Author Response · Authors · 2026-01-18
> > > > >
> > > > > By leveraging the shared embedding space of CLIP, our model exhibits a degree of semantic generalization beyond the distilled training corpus. As illustrated in Figure 9, although the word “banana” was not present in the training data, the model correctly interprets “banana color” as yellow due to the semantic proximity in the textual space.
> > > > >
> > > > > Regarding the completely unseen object categories you mentioned, it will generate 3DGS models that don‘t match the input text. It will produce shapes like humans, vehicles, and other similar forms that it can generate.
> > > > >
> > > > > However, we have demonstrated that SDS can supervise a cross-category universal generator, which was not achievable with previous baselines.
> > > > >
> > > > > Thank you for your time and constructive feedback. We stand ready to engage in further discussion and provide any additional clarifications needed.

---

> > > > > > ### Comment · Reviewer_6PBk · 2026-01-18
> > > > > > **Response to Authors' Reply**
> > > > > >
> > > > > > I appreciate the discussion with the authors, and I have made my decision.
> > > > > >
> > > > > > For future work, I hope the authors can (i) expand the category coverage to make the approach more broadly applicable, and (ii) address the over-optimization issue so the model can be used more reliably in real-world scenarios.

---

> > > > > > > ### Author Response · Authors · 2026-01-18
> > > > > > >
> > > > > > > Thank you again for your suggestions and review efforts.

---

### Review · Reviewer_SEjh · 2026-01-02

**Summary Of Contributions:**

The paper proposes BrightDreamer, a feed-forward text-to-3D generative framework that outputs a 3D Gaussian Splatting (3DGS) representation directly from a text prompt in approximately 77 ms. The key idea is to avoid directly generating millions of Gaussian parameters by (i) deforming a fixed 3D anchor grid via a Text-guided Shape Deformation (TSD) network to obtain Gaussian centers, and (ii) generating word-aware triplanes via a Text-guided Triplane Generator (TTG) and decoding per-center features into the remaining Gaussian attributes (scale, rotation, opacity, SH). Training uses SDS with a 2D diffusion model, and the resulting 3DGS can be rendered at purported 700+ FPS.

**Audience:**

No

**Audience Explanation:**

**Weaknesses**
 - Comparisons omit several close contemporaries in amortized text-to-3D and triplane-based generators (e.g., SeMv-3D, TPA3D)
- Key implementation details are missing: number of Gaussians used at inference (effective active splats), rendering resolution for FPS, training steps and wall-clock time, and Gaussian SH order.
- Several hyperparameters are specified without context (e.g., scale range (-9, -3)) and lack justification or ablation.

**Broader Impact Concerns:**

The paper presents a technical framework for accelerating text-to-3D generation. I do not see any specific ethical concerns or negative societal impacts arising from this work.

**Claims And Evidence:**

Yes

**Claims Explanation:**

- Conceptually clean decomposition: predicting Gaussian centers by deforming an anchor lattice and predicting the remaining attributes from triplanes is an intuitive way to sidestep directly regressing large Gaussian sets.
- The use of token-level cross-attention in both the TSD and TTG to inject fine-grained text guidance into geometry and appearance is reasonable and consistent with recent advances in text-conditioned 3D generation.
- The argument to avoid channel-splitting for triplanes (three separate generators) targets a concrete representational deficiency and may improve cross-plane coherence in practice.

**Requested Changes:**

**Questions for Authors**

- How many Gaussians are active at inference (on average) and what pruning strategy do you use for low-opacity splats? How does pruning affect quality and rendering FPS?
- What rendering resolution and SH order are used for the reported 705 FPS? Please report FPS as a function of resolution and Gaussian count to aid fair comparison.
- How many training steps and wall-clock time are required to train the 500M-parameter model? What GPUs are used, and what is the total compute budget?
- How large is the prompt dataset (number of prompts, categories, source), and what is the distribution of compositional vs single-object prompts? How are train/test splits defined?
- Can you report standardized quantitative metrics (e.g., CLIP scores on multi-view renders, T3Bench, multi-view consistency metrics) against ATT3D/SeMv-3D/TPA3D/Instant3D and per-prompt baselines?
- Please provide evidence for the “spatial inhomogeneity” claim: can you quantify cross-plane correspondence/coherence or show consistency metrics improved by your three-generator design?
- How sensitive are results to the displacement bound β and the anchor grid resolution (e.g., 32^3 vs 64^3 vs 128^3)? Does increasing β improve expressivity or destabilize training?
- Do you employ any regularization on opacity or sparsity losses to avoid retaining many near-zero-opacity Gaussians? If so, please detail the loss and its weight.
- What diffusion backbone and guidance scales are used in SDS during training? Did you try multi-view diffusion or image-conditioned priors (e.g., MVDream/Zero123) to improve consistency?
- How do you ensure that the word-level cross-attention in TTG/TSD scales to complex prompts (e.g., long sentences) without losing local attributes? Any failure cases or mitigation strategies?

---

> ### Author Response · Authors · 2026-01-11
>
> Sincerely, thanks for reviewing our paper and providing constructive suggestions to help us improve the paper. We have fixed the problem and made clearer statements according to your requested changes, highlighted in cyan in the revised paper. We hope our responses adequately address your concerns, and we remain available for any further discussion or clarification you may require.
>
> ## W1 Missing baselines
>
> The mentioned methods, SeMv-3D and TPA3D, are all **closed-source** methods. We can not reproduce their result for comparison.
>
> ## W2 Missing details
>
> The number of 3DGS is fixed at 64^3. The resolution for rendering speed is 512 x 512. We train our models for 30,000 iterations in two days. We only use 1-order SH coefficient (3 dimensions), the same as the majority of text-to-3D methods.
>
> ## W3 The hyper-parameter setting
>
> The setting of the scale range is based on the spatial coordinates scale. This value is set within this range to prevent the size from being too large or too small.
>
> ## Q1 3DGS prune
>
> We do not prune any 3DGS points. Since we are a feed-forward method and directly output 3DGS in a single inference, we do not need a pruning strategy in the original 3DGS.
>
> ## Q2 Rendering resolution and SH order
>
> The number of 3DGS is fixed at 64^3. The resolution for rendering speed is 512 x 512. We train our models for 30,000 iterations in two days. We only use 1-order SH coefficient (3 dimensions) as same as the majority of text-to-3D methods.
>
> ## Q3 Training cost
>
> The setting of the scale range is based on the spatial coordinates scale. This value is set within this range to prevent the size from being too large or too small.
>
> ## Q4 Training dataset
>
> The generator is trained on a single prompt set including vehicle, daily life, and animal descriptions, in a total of 30K sentences. For the test, we directly write some sentences that are not in the training set.
>
> ## Q5 CLIP score
>
> Thanks for your suggestions. We added the CLIP score comparison with Instant3D and LucidDreamer. For SeMv-3D and TPA3D, they are closed-source methods, we can not compare with them.
>
> | Method     | Instant3D | LucidDreamer |  Ours |
> |------------|:---------:|:------------:|:-----:|
> | CLIP Score |   0.322   |     0.286    | 0.324 |
>
> ## Q6 Spatial inhomogeneity
>
> As shown in Fig.4 of the paper. When the convolution kernel scans the 2D planes, if we unfold it into a 3D spatial system, we can find that it moves simultaneously on three axes. This leaves many empty areas. And our ablation study proves that we achieve a better quality after we fix this problem.
>
> ## Q7 Beta and Spatial resolution
>
> If the spatial resolution is too low, the generated results will be too coarse to accept. So we select the max spatial resolution that our GPU can support. And the bound beta is to limit the free area for each 3DGS point. If we set this value too high, the training will crack.
>
> ## Q8 Losses
>
> We do not employ any regularization on opacity or sparsity losses. Our method directly predicts all the properties of 3DGS points in a single forward inference. So we do not need to optimize each property of 3DGS points. The near-zero-opacity area illustrates that the area is empty.
>
> ## Q9 SDS backbone
>
> We use DeepFloyd IF to compute the SDS loss. We do not utilize MVDream due to the GPU memory limitation. Moreover, we do not use image-conditioned SDS since our dataset only contains text. And we focus on the text-to-3D task.
>
> ## Q10 Cross-attention
>
> Because each word is involved in the formation of the final 3DGS, the details of the sentence will not be lost. In addition, our text is not very long, mostly 10-20 words, so this phenomenon will not occur.

---

> > ### Comment · Reviewer_SEjh · 2026-01-17
> > **Feedback**
> >
> > My concerns are fully addressed

---

> > > ### Author Response · Authors · 2026-01-17
> > >
> > > Sincerely, thank you again for reviewing our paper.

---

### Decision · Action_Editor_GW6c · 2026-03-01

**Recommendation:** Accept with minor revision

**Additional Comments:**

The paper introduces a simple text-to-3D method that yields decent efficiency. The experimental validation is sufficient and solid. All reviewers suggest acceptance and I agree to accept this paper. The authors are encouraged to include all the additional results in the rebuttal to their final manuscript.

**Audience:**

Yes

**Audience Explanation:**

Text-to-3D is an important generative task in 3D vision. It is of sufficient interest for both machine learning and 3D vision people.

**Claims And Evidence:**

Yes

**Claims Explanation:**

The paper introduces a fast test-to-3D syntheis method for 3D Gaussian Splatting. The results show sufficient evidence to support the claim.